# An Underwater Human–Robot Interaction Using a Visual–Textual Model for Autonomous Underwater Vehicles

**DOI:** 10.3390/s23010197

**Published:** 2022-12-24

**Authors:** Yongji Zhang, Yu Jiang, Hong Qi, Minghao Zhao, Yuehang Wang, Kai Wang, Fenglin Wei

**Affiliations:** 1College of Computer Science and Technology, Jilin University, Changchun 130012, China; 2State Key Lab of Symbolic Computation and Knowledge Engineering of Ministry of Education, Jilin University, Changchun 130012, China

**Keywords:** autonomous underwater vehicle, underwater human–robot interaction, gesture recognition, visual–textual association

## Abstract

The marine environment presents a unique set of challenges for human–robot interaction. Communicating with gestures is a common way for interacting between the diver and autonomous underwater vehicles (AUVs). However, underwater gesture recognition is a challenging visual task for AUVs due to light refraction and wavelength color attenuation issues. Current gesture recognition methods classify the whole image directly or locate the hand position first and then classify the hand features. Among these purely visual approaches, textual information is largely ignored. This paper proposes a visual–textual model for underwater hand gesture recognition (VT-UHGR). The VT-UHGR model encodes the underwater diver’s image as visual features, the category text as textual features, and generates visual–textual features through multimodal interactions. We guide AUVs to use image–text matching for learning and inference. The proposed method achieves better performance than most existing purely visual methods on the dataset CADDY, demonstrating the effectiveness of using textual patterns for underwater gesture recognition.

## 1. Introduction

Autonomous underwater vehicles (AUVs) can effectively assist divers in complex operations and mitigate risks [1,2,3,4,5] in applications such as marine science, archaeology, and the maintenance of marine infrastructure. AUVs rely on acoustic, inertial, and visual sensors for intelligent decision-making. Figure 1 demonstrates the underwater human–robot interaction (U-HRI) platform, which is built in the CADDY [6,7] project. Specifically, the communication from the diver to the machine is recorded by the AUVs’ cameras with a backend algorithm that analyzes its semantic information. Communication from the AUVs to the diver is signaled by the AUVs’ flashing lights or through the screens to show more complex information feedback to the diver. The sonar keeps the diver and the AUVs at a distance by detecting the relative positions of the divers. However, due to the low-light underwater environment [8], equipment conditions [2], and restricted human action, U-HRI is much more complicated than land operations. Divers wear special equipment, such as rebreathers, which makes it very difficult to achieve voice communication [1]. Hand gestures are the standard way of communication between divers due to their clear semantics and simplicity of operation. Similarly, gesture recognition is widely used in underwater human–robot interactions.

Due to the distractions of the underwater environment [8,9,10] and the equipment (the overall color of the diving suit is very close to that of the background) [1], current underwater gesture recognition methods used by AUVs usually use a two-stage network architecture [11]. Although the current two-stage network structure has a better accuracy rate, these methods are limited by the efficiency and accuracy of both the detector and classifier components. On the other hand, these methods rely on presegmented images of a dataset under benign conditions. Inspired by recent vision–text-related work [12,13,14], we realized that there was an intrinsic relationship between images and texts. We tried to consider this problem in terms of the way humans learn. Humans learn by matching the appearance of objects to their names rather than directly matching images to labels. Therefore, we applied visual–text multimodal learning in this task, using text as supervised information for a limited number of underwater images.

This paper proposes an effective visual–textual framework for underwater gesture recognition. We improve the accuracy of gesture recognition by using text semantic information. Our multimodal framework consists of two independent unimodal encoders for images and labels and a multimodal interaction module. This framework extracts visual features from the underwater images and textual features from the labels to infer the meaning of the diver’s gestures after multimodal interaction. As shown in Figure 2, the training process matches the image and the label feature pairs as closely as possible, while keeping specific image features away from other mismatched labels. As a result, visual–textual learning brings AUVs closer to human behavioral habits. In the inference phase, the proposed model is a visual–textual matching problem rather than a 1-N majority vote method.

Furthermore, the lack of samples [6,7] makes it challenging to construct visual–textual learning resources. In contrast, the Internet contains many image resources rich in textual markup. Inspired by recent work [15,16,17,18], by using pretrained models as a solution, we redefined the task to bootstrap the visual–textual task and fine-tuned the entire model for the underwater dataset. Our contributions can be summarized as follows:Underwater gesture recognition is constructed as a multimodal problem, and the U-HRI performance of an AUV is optimized by introducing a text modality to fully explore the feature associations between images and text.We propose a new underwater visual–textual gesture recognition model (VT-UHGR).Extensive experiments on the public benchmark dataset CADDY [6,7] show that the proposed visual–textual approach achieves superior performance over the visual-only approach.

## 2. Related Work

### 2.1. Hand Gesture Recognition

The current mainstream deep-learning-based gesture recognition methods include appearance-based recognition [19,20], motion-based recognition [21,22], skeleton-based recognition [23,24], depth-based recognition [25,26], 3D-model-based recognition [27,28,29,30], etc. Chen et al. [19] proposed a two-level approach for real-time gesture classification, with the lower-level approach based on Haar-like features and AdaBoost learning algorithm and the upper-level approach based on a contextual syntactic analysis. Saha et al. [20] relied on the choice of an active difference signature-based feature descriptor and used the HMM model to improve the feature-based methods’ performance significantly. Simonyan et al. [21] proposed a spatiotemporal dual-flow structure for recognition using single-frame and front–back optical flow images. Zhu et al. [22] used computational implicit motion information to reduce the high computational cost associated with computing optical flow in motion-based methods. Devineau [23] used parallel convolution to process the position sequences of hand-skeletal joints and still achieved advanced performance using only hand-skeletal data for recognition. Cai et al. [28] proposed a weakly supervised method, adapting from a fully annotated synthetic dataset to a weakly labeled real-world dataset with a depth regularizer.

### 2.2. Underwater Gesture Recognition

Due to the influence of the underwater environment [8,9,10,31,32], underwater images suffer from noise interference, refraction effect, wavelength color attenuation, and other problems. Therefore, it is challenging to accurately identify diver images captured by AUVs in complex underwater environments. Current image-based deep learning methods do not directly yield good performance in underwater environments. In [33], pretrained versions of several classical network models were trained for underwater environments using a migration learning approach, including AlexNet [34], VggNet [35] ResNet [36], and GoogleNet [37]. In [38], an attempt was made to solve this problem from detection to classification using Faster R-CNN [39], deformable Faster R-CNN [11], and other efficient detectors, and a comprehensive analysis of the impact of underwater data was developed. Zhao [40] et al. deployed an underwater real-time FPGA system that carried a lightweight convolutional neural network for underwater image recognition.

### 2.3. Vision–Text Multimodality

Recent work on semantic–textual information has attracted much attention, such as pretraining [15,16], visual–textual retrieval [12,13,14], action recognition [17,41,42], attribute recognition [18], etc. CLIP [13] learned transferable visual models from natural language supervision and achieved surprising results on classification tasks. ActionClip [17] verified the effectiveness and scalability of text supervision in video-based action recognition. VTB [18] introduced text supervision to the multiclassification task of pedestrian attribute recognition and significantly improved the results. Fine-tuning on a specific dataset significantly improves the model’s performance [15,16].

## 3. Method

The current gesture recognition methods used by AUVs [21,22,23,24,25,26,27,28,29,30] are essentially the classical 1-N majority vote problem that maps the corresponding labels to numerical categories without using the rich semantic information in the labels. Instead, we model the gesture recognition task used by AUVs through an image–text multimodal representation. Specifically, we use semantic information from the text to supervise the classification of images. We take advantage of the improved cognitive approach of the AUV, thus optimizing the accuracy of gesture recognition. As shown in Figure 2, (a) is the current traditional framework and (b) is our multimodal framework (b). (a) uses an end-to-end approach or a two-stage approach to map labels into numbers or one-hot vectors, while (b) tries to pull the semantic information of the label text and the corresponding image representation close to each other.

As shown in Figure 3, we propose a visual–textual baseline as an underwater gesture recognition model (VT-UHGR), which consists of three modules, including visual feature extraction (VFE), textual feature extraction (TFE), and multimodal interaction. The visual feature extraction module generates visual features from the input diver’s image. The textual feature extraction module encodes the input label into the corresponding textual features. Furthermore, the multimodal interaction module projects features of two modalities to an identical high-dimensional semantic space and generate visual–textual features using a transformer encoder. Gesture classification predictions are generated from the corresponding visual–textual features by an independent feed-forward network (FFN). Benefiting from the representational power of the transformer, VT-UHGR fully explores intra- and cross-modal correlations. In addition, the location and modality type embeddings are used to maintain spatial and modal information. In subsequent subsections, we describe each module and the employed pretrained methods.

### 3.1. Pretraining

We fine-tuned the framework using a model that had been pretrained on a large dataset, ImageNet [43], and we adapted and recustomized the downstream task to be more like the upstream pretrained task. The traditional prompt and fine-tune approach adapts the pretrained model to the downstream classification task by attaching a new linear layer to the pretrained feature extractor. Here, we used two kinds of cues: textual and visual. We used a pretrained BERT [44] as the textual encoder and a pretrained VIT [45] as the visual encoder. Fine-tuning on a specific dataset significantly improves the model’s performance [15,16]. Moreover, since we introduced additional parameters, we needed to train on these parameters. Therefore, we retrained the assembled entire framework end-to-end on the target dataset.

### 3.2. Transformer Block

The transformer block [45,46] is a type of multiplexed unit used in our network structure. As shown in Figure 4, a transformer block consists of multihead attention, add-and-norm, and MLP modules. Our visual feature extractor, text feature extractor, and multimodal interaction modules were all made up of transformer blocks stacked in different ways. The last layer of the transformer block in the BERT [44] network used for the textual feature extractor was the feed-forward layer.

The effectiveness of the transformer structure mainly relies on the self-attention mechanism [46]. In each block, the vector goes through the multihead self-attention module to get a weighted feature vector *Z*, i.e., a query feature matrix *Q* is used to calculate its similarity to each key feature matrix *K*, and then a weighted sum of all value matrices *V* is performed:(1)$$Attention\left(Q,K,V\right)=softmax\left(\frac{QK^{T}}{\sqrt{d_{k}}}\right)V$$
where $d_{k}$ is the dimensionality of the matrix *K*. In addition, to avoid the model from overly focusing on its position when encoding the information at the current position, the transformer block uses multiple attention heads to output the encoded representation information in different subspaces, further enhancing the expressive power of the model.

### 3.3. Visual Feature Extraction

As shown in Figure 5, given an image *I* of a diver, we used the VFE module to obtain its visual features *V*. In a CNN, convolving the image directly in two dimensions is sufficient without a special preprocessing process. However, the transformer structure cannot process the image directly and needs to be chunked beforehand. Specifically, we chose the VIT [45] core process as the visual encoder after making patches, patch embedding, position embedding, and a transformer block encoder to get the image features vector *F*. That is, $F=VFE\left(I\right)\in R^{C\times H\times W}$, where C,H, and *W* represent the channel size, height, and width of *F*, respectively. To integrate with the textual features, we further plasticized *F* by extending it to one dimension in the spatial dimension, resulting in a set of visual feature vectors $V=\left[V_{1},V_{2},..V_{\tau }\right]\in R^{S\times C}$, where S=H×W. In summary, the features of the diver image *I* extracted by the visual feature extractor were defined as:(2)$$V=VFE\left(I\right)=Reshape\left(VIT\left(I\right)\right)$$

### 3.4. Textual Feature Extraction

As shown in Figure 6, given a set of labels $\lambda =\left[\gamma_{1},\gamma_{2},\ldots \gamma_{\tau }\right]$, we used the TFE module to extract the corresponding textual features *T*. All original texts (e.g., “boat”, “on”, etc.) were first assembled into statements (e.g., “This picture represents ‘boat’”, “The action of the diver is ‘on’” etc.) and then encoded them to text features by the natural language method [44,47]. We referred to some natural language processing methods to transform the text of the feature labels to match them in the vector space. Specifically, for the label γ, the word vector was first obtained by an embedding, which contained the word vector token embedding, the segment embedding to distinguish the tag, and the position embedding to encode the position. In addition, the last layer of the transformer block used in BERT was the feed-forward layer. The textual encoder was not involved in the training process. It only used the pretrained model to output the corresponding textual features. Therefore, our textual module did not burden the training considerably.

We finally obtained the textual features *T* by a pretrained BERT [44] consisting of two-way transformer blocks, i.e., $T=\left[t_{1},t_{2},\ldots t_{\tau }\right]=BERT\left(Embedding\left[\gamma_{1},\gamma_{2},\ldots \gamma_{\tau }\right]\right)$. In summary, the textual extractor extracted the label λ features defined as: (3)$$T=TFE\left(\lambda \right)=BERT\left(Embedding\left(\lambda \right)\right)$$

### 3.5. Multimodal Interaction

Multimodal interaction is essential for the exchange of information between multiple modalities. An intuitive approach to modal fusion is integrating different modalities’ features by simple operations, such as weighting or cascading. In this paper, we used the transformer encoder for a deep cross-modal fusion. As input, the encoder received a sequence of features from two modalities, textual and visual. The information from both modalities interacted in the encoder through the self-attention mechanism and eventually output multimodal features.

As shown in Figure 3, our model extracts visual features *V* and textual features *T* from two feature extractors that interact in a multimodal interaction module to finally generate visual–textual features *Z* for gesture recognition. We used the transformer encoder [45] as a multimodal interaction module to allow unrestricted interaction between visual–textual sequences. On the other hand, it allowed the framework to model the correlations within and across modalities deeply. We first mapped *V* and *T* to the same higher-dimensional space *D* for sequence concatenation, as follows: (4)$$V_{\varsigma }=\left[\Phi \left(V_{1}\right)+e_{1}^{pos},\Phi \left(V_{2}\right)+e_{2}^{pos},\ldots \Phi \left(V_{\tau }\right)+e_{s}^{pos}\right]$$
(5)$$T_{\varsigma }=\left[\varphi \left(t_{1}\right),\varphi \left(t_{2}\right),\ldots \varphi \left(t_{\tau }\right)\right],$$
where $\phi \left(.\right)$ and $\varphi \left(.\right)$ are both linear 1×1 convolution layers, $E^{pos}=\left[e_{1}^{pos},e_{2}^{pos},\ldots e_{s}^{pos}\right]\in R^{M\times D}$ is the learnable location embedding, which needs to add the spatial information of the image as visual features to the extracted features. The final obtained visual–textual feature pair $Z_{0}$ was as follows: (6)$$Z^{0}=\left[V_{\varsigma }+e_{v}^{type};T_{\varsigma }+e_{t}^{type}\right]$$

The learnable modal type embedding $e_{v}^{type},e_{t}^{type}$ was added with the corresponding sequences to maintain the different modal information. Next, the initial visual–textual vectors were learned by a transformer encoder. The transformer encoder consisted of a stack of *L* transformer blocks, each of which included a multiheaded self-attention (MSA) layer and a multilayer perceptron (MLP) layer, as well as normalization layers (LN); details are shown in Figure 4. The final gesture classification results were generated by feature comparisons extracted from pretrained text labels. Specifically, we derived the corresponding predicted values by an independent feed-forward network (FFN) containing linear layers. The network used the same text features for the same set of different classified diver images.

### 3.6. Loss Function

Given a set of attribute annotations and a training dataset, we use the cross-entropy method, widely used for multiclassification tasks. Our loss function was formulated as follows:(7)$$L=-\frac{1}{N}\sum_{i}^{n}\sum_{j}^{M}\left(y_{ij}log\left(p_{ij}\right)\right)$$

In the inference phase, all text labels were encoded as text features by the TFE module, and the corresponding textual features were fed to the multimodal interaction module for prediction. The proposed visual–textual method was trained by optimizing *L* end-to-end, except for the TFE module, which was not involved in model training. Since our training goal was to make the feature vectors of an image–text pair as similar as possible, rather than a pair as similar as possible, here, the similarity was calculated using the inner vector product. In this way, our loss function was computed by *N* positive samples and $N^{2}-N$ negative samples.

## 4. Results

### 4.1. Datasets

BUDDY-AUV collected the CADDY [6,7] data used in this paper, and the University of Zagreb designed it. It is equipped with navigation sensors, a Doppler velocity logger (DVL), an ultrashort baseline (USBL), and perception sensors, including a multibeam sonar and a stereo camera in the underwater housing. Additional underwater housing enabling two-way human–machine interaction is also included.

The underwater robot captured the gesture image dataset in eight underwater scenes and the dataset contains over 10,000 images. For different scenes, each image contains the image number, shooting scene, gesture name meaning, gesture number, left-hand position, right-hand position, semantics, and other information. As shown in Figure 7, the AUV intelligently senses divers’ gestures in the underwater environment. The dataset contains common underwater communication gestures such as up dive, carry, digit, stay, and boat.

We evaluated the performance of the model using accuracy, i.e., the number of samples correctly predicted as a percentage of the total. The CADDY dataset has a relatively homogeneous sample distribution, and using accuracy gave a more intuitive picture of the model’s performance level.

In addition, to test the performance of our model in a more general scenario, we additionally chose to supplement it with the SCUBANet dataset [48], which was recorded by the Milton unmanned underwater vehicle (UUV). As shown in Figure 8, it contains two distinct scenarios of cold and warm water. Since the SCUBANet dataset is primarily designed to detect the body parts of divers, we filtered the dataset by explicit gesture semantics.

### 4.2. Implementation Details

We implemented VIT [45] as our visual coder, which was pretrained on ImageNet [43]. Correspondingly, we used a pretrained BERT [44] as a text encoder. The cell structure in the transformer encoder for multimodal interaction was the same as that of the transformer block in VIT. The dimensionality of the visual–textual features was set to 256. The images used in the training process were enhanced with random level flipping and random cropping. We used the Adam optimizer to optimize our model with $\beta_{1}=0.9,\beta_{2}=0.999$. We used a warm-up strategy to increase the learning rate linearly from 0 to an initial learning rate of $1\times 10^{-3}$ in the first ten epochs and decreased the learning rate by a factor of 0.1 when the number of iterations increased. The batch size was set to 32.

### 4.3. Comparison with the State of the Art

We compared our method with state-of-the-art end-to-end methods for underwater gesture recognition, as shown in Table 1. We compared the model’s accuracy, the number of parameters, and the average time of inference for diver gesture recognition with different methods on the same test set. The data for the comparison methods were mainly from [33]. Experiments showed that our method significantly outperformed the end-to-end methods alone in terms of accuracy. Since the sample distribution of the CADDY dataset was relative, using the accuracy rate could reflect the performance level of the model intuitively. In addition, due to the larger number of transformer structural parameters, our inference time in the same environment was slightly longer.

We compared our method with end-to-end underwater gesture recognition methods on the SCUBANet dataset, as shown in Table 2. All models showed a significant decrease in efficiency compared to the results on the CADDY dataset. The divers in SCUBANet did not wear markers on their hands, and problems such as complex backgrounds and motion blur in the dataset posed challenges for the recognition. In addition, due to the large image size of SCUBANet samples, the large scaling to fit the input of the model also posed difficulties for feature extraction.

We compared our method with state-of-the-art two-stage underwater gesture recognition methods, as shown in Table 3. We compared the accuracy of gesture recognition for divers with different methods on the same test set. The data for the comparison methods were mainly from [38]. Experiments showed that our method had comparable performance in terms of accuracy compared to the two-stage method. Our model could achieve similar performance to detection-based methods without relying on detection labeling pairs.

The superior performance of our approach was mainly attributed to the textual features provided by the pretrained textual encoder, which helped the model learn the intrinsic feature correlation between pictures and text. On the other hand, we effectively interacted with intermodal features through the multimodal interaction module. Thus, our approach was significantly better than the visual-only based approach.

### 4.4. Ablation Study

In this subsection, we conducted an extensive ablation study in the CADDY dataset. As shown in Table 4, we explored the impact of different textual encoders and the absence of textual encoders on the performance of our model. Using a text feature extractor significantly improved the performance of the visual-only model, and even when using a simple one-hot encoding approach to interact with visual features, the performance was significantly better than the visual-only scheme. In addition, more detailed processing of textual features using BERT [44] could further improve the model results. Our approach treated underwater gesture recognition as a multimodal task and fully used the textual information inherent in the labels.

As shown in Table 5, we tested the need for multimodal interaction. Specifically, we constructed a more straightforward visual–textual baseline using additive operations as a cross-modal fusion module instead of a transformer encoder. In addition, we tested the corresponding performance degradation without positional encoding and modal encoding, respectively.

Experiments showed that using the transformer block as a multimodal interaction module significantly improved performance. In addition, it was necessary to provide more numerous conditions, such as positional encoding for the encoded information. On the other hand, the introduction of the learnable modal encoding also improved the performance of the final result.

## 5. Limitation

There is still a lack of large-scale, scenario-rich, high-resolution benchmark datasets due to the complex underwater environment and the limitations of the hardware quality of the filming equipment. In addition, the intensive use of transformer blocks in the inference process places an additional burden on the hardware side of the computational deployment. Therefore, using lightweight models to achieve more accurate recognition will make more sense in the future of edge computing.

## 6. Conclusions

In this paper, we formulated underwater gesture recognition as a multimodal problem and proposed a visual–textual baseline (VT-UHGR) to improve the performance of U-HRI for AUVs. We explored the correlation between visual and textual features using the transformer block. Extensive experiments on a widely used CADDY dataset demonstrated the importance of introducing textual modalities, and our proposed visual–textual baseline achieved a higher performance than the purely visual approach.

## Figures and Tables

**Figure 1 sensors-23-00197-f001:**
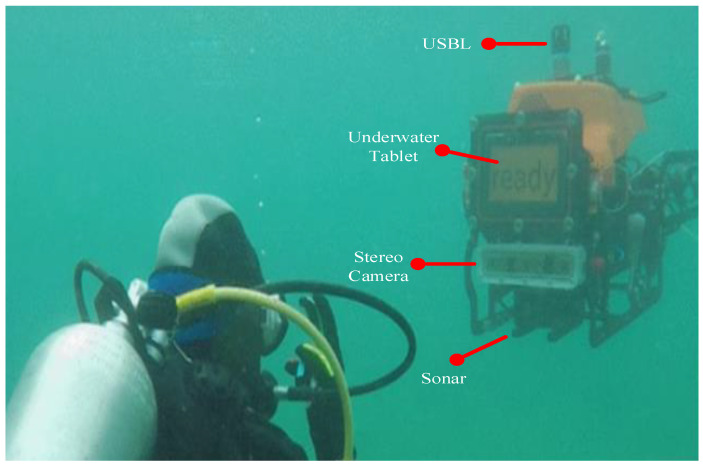
The CADDY system for assistance in diver missions as an example of U-HRI [6,7].

**Figure 2 sensors-23-00197-f002:**
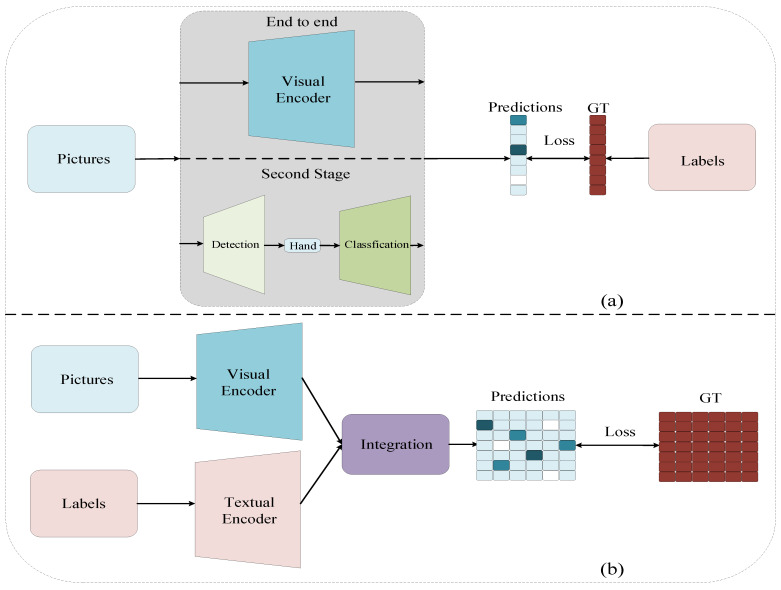
The current traditiona l framework (**a**) and our multimodal framework (**b**).

**Figure 3 sensors-23-00197-f003:**
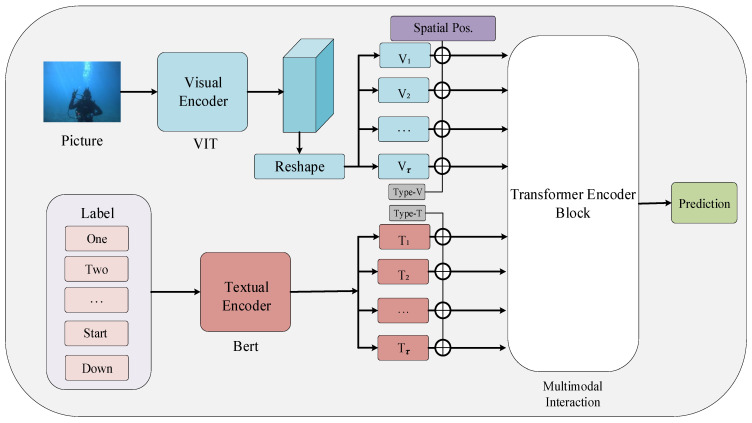
An overview of our proposed visual-textual baseline (VT-UHGR).

**Figure 4 sensors-23-00197-f004:**
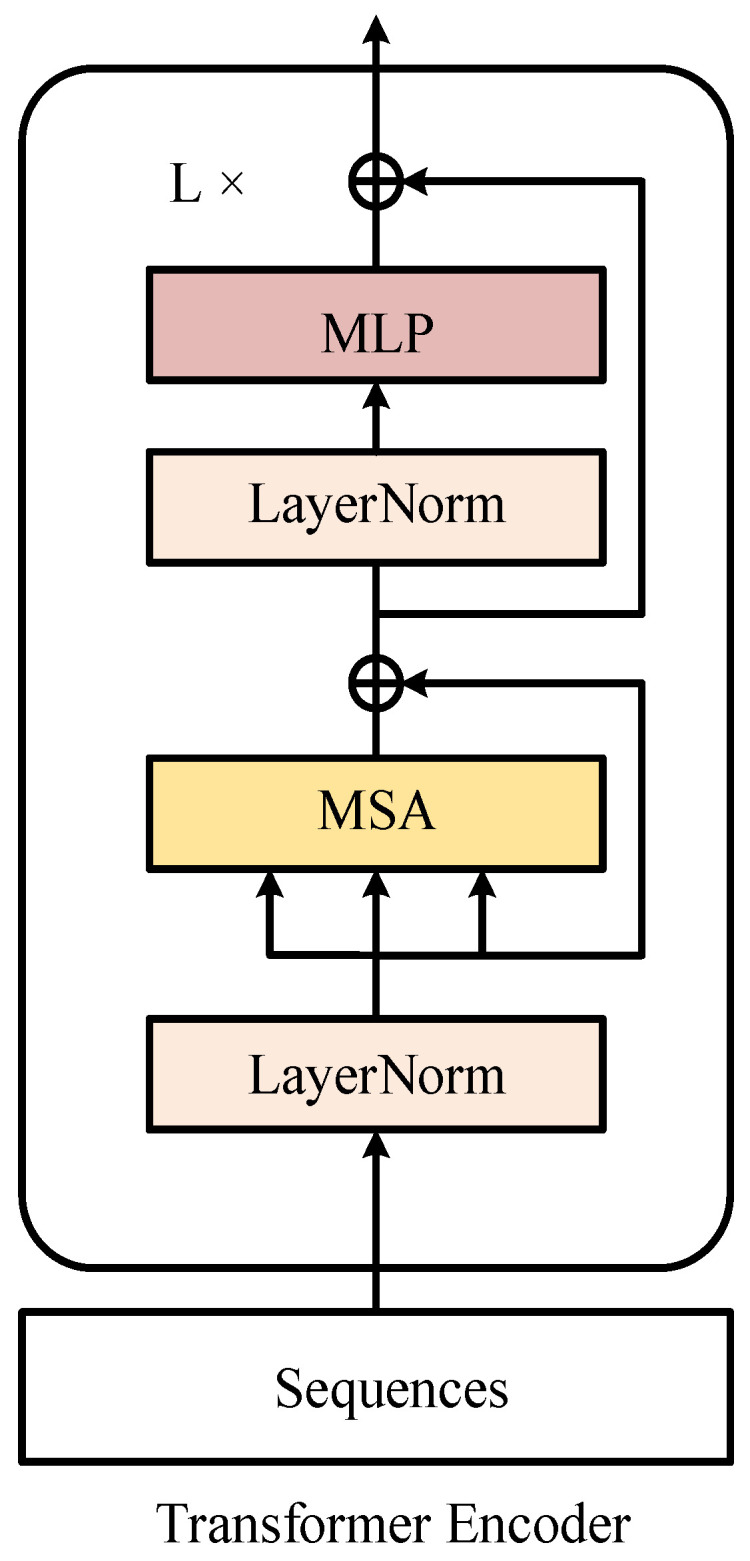
The overall structure of the transformer block.

**Figure 5 sensors-23-00197-f005:**
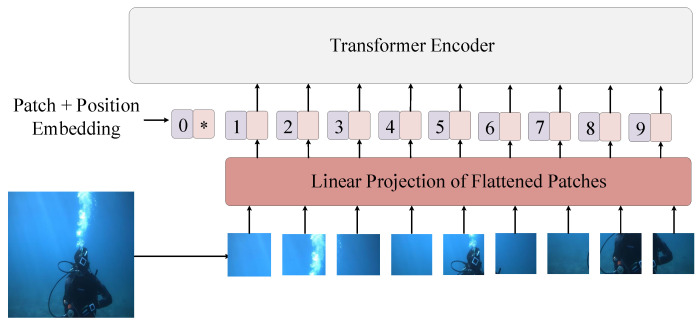
The overall structure of the VIT model.

**Figure 6 sensors-23-00197-f006:**
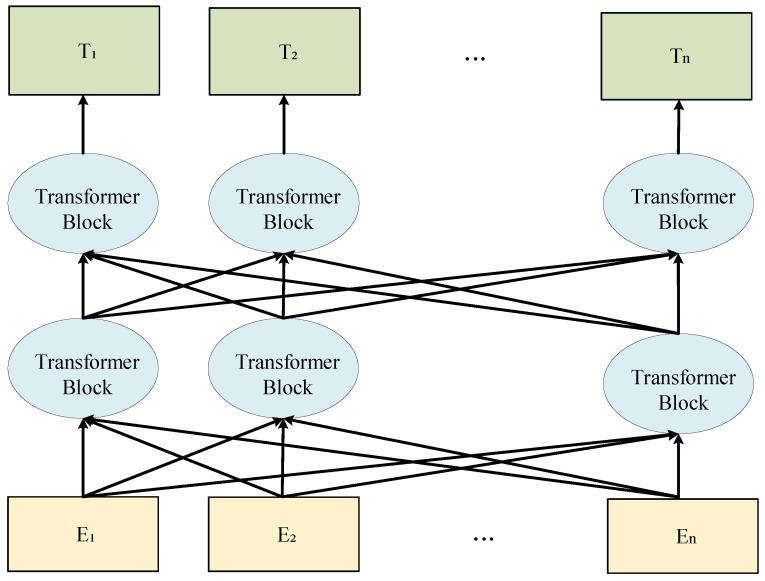
The overall structure of the BERT model.

**Figure 7 sensors-23-00197-f007:**
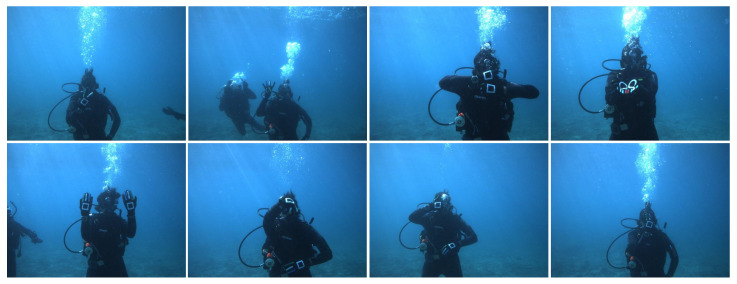
Example images from the CADDY [6,7] dataset.

**Figure 8 sensors-23-00197-f008:**
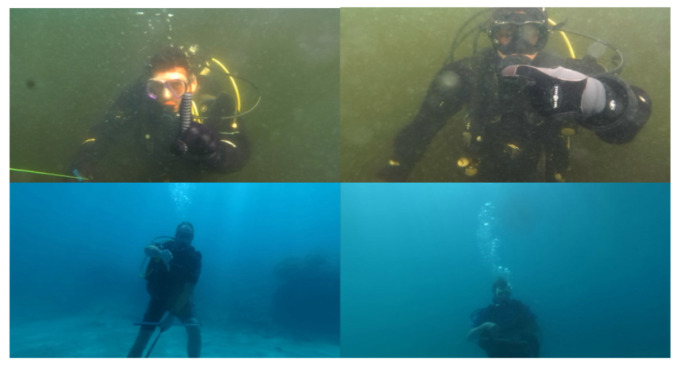
Example images of the SCUBANet [48] dataset.

**Table 1 sensors-23-00197-t001:** Quantitative comparison of end-to-end methods on the CADDY dataset.

Methods	Acc	Params (M)	Times (ms)
AlexNet	0.83	61.1	0.84
ResNet	0.88	11.7	1.26
GoogleNet	0.90	6.8	1.65
VggNet	0.95	138.4	2.14
VT-UHGR (ours)	0.98	178.4	2.87

**Table 2 sensors-23-00197-t002:** Quantitative comparison of end-to-end methods on the SCUBANet dataset.

Methods	Acc
ResNet	0.75
GoogleNet	0.78
VggNet	0.82
VT-UHGR (ours)	0.86

**Table 3 sensors-23-00197-t003:** Quantitative comparison of two-stage methods on the CADDY dataset.

Methods	Acc
MD-NCMF	0.77
SSD with MobileNets	0.85
FC-CNN with ResNet-50	0.95
Deformable Faster R-CNN	0.98
VT-UHGR (ours)	0.98

**Table 4 sensors-23-00197-t004:** Impact of different textual encoders on VT-UHGR performance on the CADDY dataset.

Methods	Textual Encoder	Acc
VIT	−	95.81
VT-UHGR	One-hot	97.13
VT-UHGR	BERT	98.32

**Table 5 sensors-23-00197-t005:** Ablation study of the proposed components in VT-UHGR on the CADDY dataset.

Methods	Structure	Acc
VT-UHGR	−	97.19
VT-UHGR	+ Transformer Encoder	97.78
VT-UHGR	+ $E^{pos}$	98.11
VT-UHGR	+ $E^{type}$	98.32

## Data Availability

Not applicable.

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
