# Peer review of "An Underwater Human–Robot Interaction Using a Visual–Textual Model for Autonomous Underwater Vehicles"

_sensors, 2022, doi:10.3390/s23010197_

Round 1

Reviewer 1 Report

I suggest you title the last part of the article as "summary"/"conclusions"

"Author Contributions" should be a separate section/chapter

 The authors focused on communication in one direction: from the diver to the robot. I would expect at least one paragraph of commentary, e.g. in a summary of how the proposed approach can be used in robot-to-diver communication.

Reviewer 2 Report

This paper proposes an visual-textual model for underwater hand gesture recognition (VT-UHGR). The VT-UHGR model encodes the underwater diver’s image as visual features, the category text as textual features, and generates visual-textual features through multimodal interactions. The following points need to be carefully considered during the revision. 

-The datasets taken for the study are very minimum. The authors are advised to considered all the possible combinations of hand gesture. 

- The results discussion requires all other parameters also not only the accuracy. The loss function results examination need to be presented.

- The validation of the proposed method requires much more analysis with different data sets. Moreover the comparative study is weak. 

Round 2

Reviewer 2 Report

The authors addressed my concerns. Paper can be accepted for publications